# Emerging Trends in Lipid-Based Vaccine Delivery: A Special Focus on Developmental Strategies, Fabrication Methods, and Applications

**DOI:** 10.3390/vaccines11030661

**Published:** 2023-03-15

**Authors:** Bharathi Karunakaran, Raghav Gupta, Pranav Patel, Sagar Salave, Amit Sharma, Dhruv Desai, Derajram Benival, Nagavendra Kommineni

**Affiliations:** 1National Institute of Pharmaceutical Education and Research (NIPER), Ahmedabad 382355, India; 2School of Veterinary Medicine, University of Pennsylvania, Philadelphia, PA 19104, USA; 3Center for Biomedical Research, Population Council, New York, NY 10065, USA

**Keywords:** vaccine, liposomes, antigen, mRNA vaccine, DNA vaccine, lipid-based carriers

## Abstract

Lipid-based vaccine delivery systems such as the conventional liposomes, virosomes, bilosomes, vesosomes, pH-fusogenic liposomes, transferosomes, immuno-liposomes, ethosomes, and lipid nanoparticles have gained a remarkable interest in vaccine delivery due to their ability to render antigens in vesicular structures, that in turn prevents its enzymatic degradation in vivo. The particulate form of lipid-based nanocarriers confers immunostimulatory potential, making them ideal antigen carriers. Facilitation in the uptake of antigen-loaded nanocarriers, by the antigen-presenting cells and its subsequent presentation through the major histocompatibility complex molecules, leads to the activation of a cascade of immune responses. Further, such nanocarriers can be tailored to achieve the desired characteristics such as charge, size, size distribution, entrapment, and site-specificity through modifications in the composition of lipids and the selection of the appropriate method of preparation. This ultimately adds to its versatility as an effective vaccine delivery carrier. The current review focuses on the various lipid-based carriers that have been investigated to date as potential vaccine delivery systems, the factors that affect their efficacy, and their various methods of preparation. The emerging trends in lipid-based mRNA vaccines and lipid-based DNA vaccines have also been summarized.

## 1. Introduction

Conventional vaccines rely on the use of killed or live-attenuated pathogens that are associated with numerous limitations such as the risk of infection, unwanted side-effects including allergic responses, instability of the biological component, and the lack of feasibility of mass in vitro production of pathogens. Therefore, subunit vaccines comprising only minimal components of the pathogens such as highly purified peptide antigens, proteins, or recombinant DNA are emerging as frontiers in vaccine research owing to their low cost of production, easy scalability, and strong safety profile [1]. However, the lack of pathogen-associated molecular patterns (PAMPs) confers poor immunogenicity to subunit vaccines. In addition, the rapid degradation of the peptides in vivo tends to reduce the effectiveness of the administered dose, thereby necessitating the development of nano-technology-based delivery carriers to overcome the aforementioned limitations. Liposomes are one of the commonly investigated nanocarriers due to their ability to cause the spatial and temporal release of the biologically active agent. The discovery of liposomes dates back to the mid-1960s, when Bangham and his co-workers observed the transformation of phospholipids, upon hydration, into self-assembled lipid bilayer vesicular structures, comprising an inner aqueous compartment [2]. Over the course of yaears, liposomes have gained remarkable attention for their applicability as drug delivery carriers, owing to their biocompatibility, biodegradability, controlled-release characteristics, and lower toxic effects. The versatility of liposomes can be attributed to the fact that hydrophilic components can be incorporated within the inner aqueous compartment, whereas hydrophobic components can be loaded in the lipidic bilayer [3]. In recent years, liposomes and other lipid-based nanocarriers such as virosomes, bilosomes, vesosomes, pH-fusogenic liposomes, transferosomes, immuno-liposomes, ethosomes, and lipid nanoparticles are being widely investigated as vaccine delivery carriers owing to their ability to render protein antigens in a particulate form, that prevents its enzymatic degradation by the host cells, thereby lengthening the in vivo half-life. Further, the immunostimulatory potential of lipid-based nanocarriers makes them ideal for effective vaccine delivery. The particulate nature of such carriers facilitates their uptake by the antigen-presenting cells, further allowing the processing and presentation of the incorporated antigen through the major histocompatibility complex molecules. This in turn results in the activation of the adaptive immune system [4]. Further, the ability to attain the desired characteristics such as size, charge, size distribution, entrapment, and site-specificity through modifications in the lipid composition and the method of the preparation adds to its versatility as an effective vaccine delivery carrier [5]. Synthetic mRNAs encoding antigens, upon direct administration, have demonstrated the safety and induction of immune-responses in several clinical trials. However, in comparison to the immune responses elicited by naked mRNAs, nanocarrier-based formulations are expected to provide more specificity and internalization into the dendritic cells, enabling dose reduction [6]. Lipid-based mRNA vaccines demonstrated promising outcomes in the prevention of severe acute respiratory syndrome coronavirus 2 (SARS-CoV-2) infection, improving the lives of millions of people, in times of a global pandemic. Moreover, lipid carriers allowed the development and large-scale production of mRNA vaccines at such a fast pace, that has never been achieved in history. mRNA-1273 and BNT162b2 developed by Moderna and Pfizer-BioNTech, respectively, are great examples of such lipid nanoparticle-encapsulated mRNA vaccines that have exhibited an efficacy of around 95% (BNT162b2) and 94.1% (mRNA-1273) in clinical trials [7,8]. The aim and scope of this review is to discuss in detail the various types of lipid-based nanocarriers used in vaccine delivery and the mechanisms by which it activates immune responses. Further, the review gives a comprehensive account of the emerging trends and recent developmental strategies on lipid-based vaccine carriers and their role as vaccine adjuvants.

## 2. Mechanism of Immunostimulatory Activity of Lipid-Based Nanocarriers

The innate immune system acts as the first-line host defense mechanism that rapidly recognizes and responds against foreign pathogens. It comprises proteins of the complement system and phagocytic cells such as the macrophages and dendritic cells, that express pattern recognition receptors (PRR). PRR triggers the immune response upon recognition of PAMPs located on pathogens. In particular, the Toll-like receptors (TLRs), a sub-class of PRR, expressed on the surface of antigen-presenting cells (dendritic cells and macrophages) play a crucial role in the recognition of PAMPs and the subsequent activation of the immune responses [9]. The second line of defense, called adaptive immunity is exerted through specific immune responses to molecular determinants of pathogens. These responses are triggered as a result of the activation of the T-lymphocytes that comprise CD4^+^ T-helper cells and CD8^+^ T-helper cells and B-lymphocytes [5]. 

Lipid-based nanocarriers are recognized and internalized by the antigen-presenting cells localized in peripheral tissues through phagocytosis or receptor-mediated endocytosis. The size of nanoparticles determines the mechanism of uptake which further explains the size-dependent immunogenicity of particles. Nanocarriers less than 150 nm are taken up by clathrin-mediated endocytosis, whereas particles in the size range of micrometers undergo phagocytosis [10]. The phospholipids that constitute the nanocarriers play a crucial role in the initiation of innate immune responses. In order to obtain the desired immune response, the composition of the nanocarriers can be tailored through modification of the physicochemical properties of the vesicles such as size, charge, and type of phospholipid, and surface modifications such as attachment of a targeting moiety (Figure 1). Cationic lipids are found to be more effective vaccine adjuvants in comparison to anionic and zwitter ionic lipids owing to enhanced electrostatic interaction between the cationic lipid-based nanocarrier and the negatively charged moieties on the surface of antigen-presenting cells [11]. This in turn facilitates the fusion of nanocarriers, followed by its cellular internalization and subsequent release of the antigen payload. The immunostimulatory effects of a wide range of cationic lipid carriers have been reported, the most common ones being cationic lipids such as dimethyl dioctyldecylammonium and 1,2-dioleoyl-3-trimethylammonium-propane (DOTAP) [12]. In general, following recognition, fusion, and cellular uptake, the antigens are processed in the antigen-presenting cells and presented to MHC class I or class II molecules. MHC class II presentation causes activation of the T-helper cells, thus stimulating antibody production/cellular immunity. Exogenous pathogens or toxins are usually presented to MHC class II molecules, whereas endogenous pathogens are presented to MHC class I molecules. However, the induction of immune response through vaccination necessitates the presentation of exogenous antigens to MHC class I molecules. This is known as cross-presentation which eventually leads to the stimulation of cytotoxic T-lymphocytes [1]. Further, lipid-based nanocarriers are also found to exhibit the ‘depot effect’ that allows retention of the antigen at the site of injection, thereby enhancing the time of vaccine exposure to cells of the immune system [13]. 

## 3. Types of Lipid-Based Nanocarriers for Vaccine Delivery

### 3.1. Liposomes

Conventional liposomes have exhibited excellent potential for the delivery of vaccines due to their properties such as biodegradability, biocompatibility, and the versatility to load both hydrophilic and lipophilic components [14]. The size of liposomes typically ranges from 20 nm to a few micrometers in diameter with the phospholipid bilayer being around 4 to 5 nm thick [15]. The first ever liposome-based commercial product Doxil^®^ comprising the therapeutically active agent, doxorubicin hydrochloride, was approved for the treatment of AIDS-related Kaposi’s sarcoma by the USFDA in 1995. The use of Doxil^®^ for around 20 years has been successful due to its superiority over conventional doxorubicin formulations in terms of its unique pharmacokinetics and bio-distribution. This can be attributed to the enhanced permeation and retention effect of liposomes. The resultant lowering of side effects, in particular, marked by a drastic decrease in doxorubicin-associated cardiotoxicity along with enhanced circulation half-life of the pegylated-liposomes has improved compliance and the quality of life of patients to a remarkable extent [16]. This eventually paved way for the approval of several liposome-based products that are currently marketed for various indications including neoplastic meningitis (Depocyt^®^), acute lymphoblastic leukemia (Marqibo^®^), and severe fungal infections (Ambisome^®^) [17]. In terms of vaccine delivery, liposomes can provide controlled release of the encapsulated antigens and are amenable to a wide range of chemical modifications that enable the attachment of targeting ligands to the surface. Further, the ease of modifying its physicochemical properties such as size, surface charge, and lipid composition has made liposomes a popular nanocarrier in vaccine delivery. The efficiency of antigen loading, apart from the choice of liposomal preparation methods and physicochemical properties of the bilayer, is also dependent on certain characteristics of the antigen such as the polarity and partition coefficient (Log P) [14]. In general, drugs with a negative Log P value are highly hydrophilic and are encapsulated in the aqueous compartment of the liposome, whereas drugs with a high Log P value are lipophilic and embedded in the lipidic bilayer. Hydrophilic components are generally not encapsulated with high efficiency and demonstrate leakage owing to their lesser affinity towards the lipid phase. Several modifications have been attempted to improve the affinity of hydrophilic components that includes conjugation with polymers, pairing with surfactants that carry an opposite charge, and the use of lipophilic prodrugs [18]. In the case of vaccine delivery systems, the antigens can be embedded within the lipid bilayer or encapsulated in the aqueous core, or covalently conjugated/adsorbed on the surface of liposomes. The location of antigens in the liposomes has a profound impact on the subsequent processing and presentation pathway. Surface-adsorbed antigens are recognized easily by the B-cells, while antigens that are encapsulated within the liposomes, need to be degraded before it is available for recognition by the immune system. A comparative study between encapsulated antigens and surface-adsorbed antigens demonstrated that haemagglutinin adsorbed onto the surface of the liposomes was more immunogenic than the ones encapsulated within the liposomes [14]. However, despite the numerous advantages of conventional liposomes, the lack of stability of liposomes in vivo is an issue of major concern, that results in bilayer rupture, leakage, and pre-mature release of the antigens. Further, the shelf-life of a liposomal product is dependent on its physical stability and the propensity of the phospholipids to undergo chemical degradation upon storage. Leakage of the encapsulated antigens due to enhanced permeability of the lipid bilayer, change in particle size distribution and hydrolytic or oxidative cleavage of the phospholipids are the instability problems that need to be taken care of. However, recently, several approaches have been utilized to overcome the stability issues of liposomes. The use of saturated lipids instead of unsaturated ones reduces the level of oxidizable lipid groups in the membrane, thereby conferring protection to the vesicles against oxidation. Storage of the liposomal dispersion in its dry form, by freeze-drying (lyophilization), enables us to overcome the issues related to oxidation and hydrolysis of the product during its shelf-life. Incorporation of cryoprotectants such as sucrose, lactose and trehalose protect the vesicles against membrane rupture/fracture, thereby preventing any loss of encapsulated material or changes in the size distribution [19]. The biological stability of liposomes can be enhanced by reducing the propensity of liposomes to undergo rapid uptake by the cells of the mononuclear phagocytic system. Several approaches such as complexation between the liposomal vesicles and polymers and grafting of hydrophilic polymers such as poly (ethylene glycol) have been investigated to produce sterically stabilized liposomes [20]. Over time, the newer generation of lipid-derived nanocarriers is emerging at a rapid pace which has overcome the limitations of conventional liposomes and has demonstrated enhanced characteristics. The section below discusses the potential role of such nanocarriers in effective vaccine delivery. 

### 3.2. Virosomes

Virosomes are immunological adjuvants that were first produced by the relocation of the protein projections of the influenza virus, namely haemagglutinin and neuraminidase on the surface of unilamellar liposomes. The subunit hemagglutinin promotes virus binding, membrane fusion, and subsequent internalization into the host cells [21], whereas the enzyme neuraminidase promotes the cleavage of α-ketosidic linkage between sialic acid and the adjacent sugar residue in mucins, thereby lowering the viscosity of the mucous. This aids the virus to access the epithelial cells and further facilitates its mobility to and from the site of infection. Neuraminidase is also found to induce the production of antibodies [22]. On the isolation of the protein projections from the virus particles and subsequent purification, attachment of the subunits to the liposomes was found to occur through intercalation of the hydrophobic regions of the subunit into the lipid layer. The resulting structures were stabilized by van der Waal’s forces, and further, upon the addition of anti-influenza serum, aggregation of the viral liposomes indicated the availability of most of the binding sites for antibody attachment [23]. The virosomes thereby serve as a promising adjuvant in overcoming certain limitations associated with inactivated influenza virus vaccines such as pyrogenicity. Being both non-pyrogenic and immunogenic, virosomes have carved a niche in vaccine research with several commercial vaccines based on virosomal systems available today. Epaxal^®^, an aluminum-free virosomal vaccine, is based on the adsorption of formalin-inactivated hepatitis A virus (HAV) on to the surface of virosomes [24]. Epaxal^®^ is highly immunogenic where the administration of two doses 12 months apart gives real-time protection of at least 11 years and is predicted to last for at least 30 years in more than 95% of individuals. The ability to elicit strong antibody titers, within several days following immunization and the long-term protection conferred by Epaxal^®^, are the major factors responsible for its commercial success [25]. Inflexal^®^ V, a virosomal adjuvanted influenza vaccine has shown an excellent tolerability profile during the past two decades on the market owing to its biocompatibility and purity. The vaccine does not contain thiomersal or formaldehyde making it completely biodegradable. The low ovalbumin content which is an indication of the amount of residual egg protein reflects the purity of the vaccine. Further, it has demonstrated a remarkable humoral immune response as per the European Medicines Agency’s immunogenicity criteria that is based on seroprotection, geometric mean titer fold increase, and seroconversion [26]. Strategies such as the immune stimulating complex technology have been investigated widely for their dose-sparing potential. The majority of humans are immunologically naïve to avian influenza viruses such as H5N1. Therefore, two doses of vaccine are required to generate an acceptable antibody response. Formulating virosomal H5N1 vaccines with effective adjuvants enables the stimulation of immune responses at an antigen dose that is comparatively lower than that of the currently available seasonal influenza vaccines. Matrix MTM is one such adjuvant formed through the binding of cholesterol to saponins. It contains a mixture of two saponins, Matrix C, and Matrix A. The former is a highly reactogenic saponin whilst the latter is a well-tolerated but weaker adjuvant. Phase-1 clinical trial in healthy adults showed that the virosomal vaccine formulated with the adjuvant Matrix MTM enhanced the antibody response enabling a significant dose-sparing down to 1.5 μg of haemagglutinin, as compared to 30 μg of haemagglutinin in the non-adjuvanted virosomal vaccine alone [27]. The simplicity in production and industrial scalability of virosomes allows the manufacturing of adequately large batch sizes containing up to 500,000 doses as standard. Hence, virosomes can be considered one of the most dynamic and successful lipid-based nanocarriers in the field of modern vaccinology [28]. Figure 2 represents the image of an influenza virosome. 

### 3.3. Bilosomes

Bilosomes are vesicular structures composed of various lipids and non-ionic surfactants that are enriched by bile salts to prevent membrane destabilization thereby protecting the bioactive from degradation in the gut microenvironment. Bilosomes are prepared by incorporating lipids such as cholesterol, distearyl phosphatidyl ethanolamine (DSPE), soybean phosphatidylcholine (SPC), dipalmitoyl phosphatidylethanolamine (DPPE), and surfactants such as sorbitan monooleate and sodium tetradecyl sulfate in a combination of varying ratios. Further, stabilization of the vesicles is achieved through the addition of different bile salts such as sodium taurocholate, sodium taurodeoxycholate, chenodeoxycholic acid, and sodium glycocholate [29]. In recent years, bilosomes have been extensively explored as ideal carriers for effective oral vaccine delivery owing to their numerous advantages over conventional liposomes. Mostly, the marketed vaccines available are given through the parenteral route of administration thereby eliciting a systemic immune response. However, a systemic immune response fails to confer protection at the mucosal level which acts as the major site of entry of infectious pathogens [30]. The antigens encapsulated in bilosomes escape degradation on exposure to the harsh pH conditions of the gastrointestinal environment (Figure 3). Further, bilosomes facilitate the transport of poorly permeable proteins and polypeptides across the mucosal epithelium and efficiently deliver the antigens to macrophages and dendritic cells thereby stimulating both mucosal and systemic immunity [31]. Premanand et al. investigated the potential of recombinant baculovirus-VP1-loaded bilosomes as an oral vaccine against human enterovirus 71 (HEV71), a pathogen that causes hand, foot, and mouth disease [32]. Oral delivery of the bilosomes encapsulating tetanus toxoid demonstrated the induction of both mucosal and systemic immunity against a bacterial protein antigen [33]. Bilosomes loaded with Hepatitis B antigen (HBsAg) were developed by Shukla et al. by utilizing sorbitan tristearate, cholesterol, and diacetyl phosphate in the ratio of 7:3:1 enriched by 100 mg of sodium deoxycholate (SDC) which demonstrated an entrapment efficiency of 18–22%. The images of fluorescence microscopy revealed the selective localization and subsequent uptake of bilosomes by the gut-associated lymphoid tissues (GALT) specifically through the Payer’s patches. Further, high-dose HBsAg bilosomes containing 50 μg of the antigen demonstrated comparable anti-HBsAg IgG levels in the serum to those observed following the intramuscular administration of alum-adsorbed HBsAg (10 μg). Although a five times higher dose is required for the bilosomes to produce comparable IgG serum titers to that of intramuscular administration, it is noteworthy that intramuscular immunization failed to elicit immune responses in mucosal secretions. However, the bilosomes elicited measurable IgA levels in the mucosal secretions thereby establishing the ability of bilosomes to induce both systemic and mucosal immune responses [34]. Similar studies have demonstrated the elicitation of the mucosal immune response characterized by enhancement in the production of IgA antibodies with a range of antigens including tetanus toxoid [33], diphtheria toxoid [31], and A/Panama influenza haemagglutinin antigen [35]. Therefore, vaccine delivery systems such as the bilosomes can be a needle-free and painless immunization approach that could consequently increase patient compliance and vaccine coverage.

### 3.4. Vesosomes

The unilamellar liposomes tend to cause premature release of contents in the physiological environment due to enzymatic degradation. To overcome the aforementioned limitation, multivesicular bodies termed ‘vesosomes’ were developed by Boyer et al. that involved encapsulating unilamellar liposomes within a second bilayer for maximizing the retention of the vesicular contents. The exterior membrane serves as a physical barrier thereby protecting the contents of the internal compartment from the action of lipases. Further, upon exposure to the complex environment of the serum, a reduction in the drug release by two orders of magnitude was observed thus enhancing the circulation time of the antigen [36]. Vesosomes have been explored as effective carriers for transcutaneous immunization, a novel strategy that involves the application of adjuvanted antigens onto the hydrated bare skin. The dense distribution of antigen-presenting cells and the immunocompetent Langerhans cells in the skin enables the production of robust immune responses upon topical administration of the vaccine formulation. Mishra et al. developed novel fusogenic vesosomes for the potential delivery of tetanus toxoid through the topical route. The developed system comprised inner cationic liposomes with a lipid composition of phosphatidylcholine, dioleoyl phosphatidylethanolamine, and dioleoyl trimethyl ammonium propane in the molar ratio of 2:1:0.5 that was, in turn, encapsulated in an outer liposomal bilayer containing dipalmitoyl phosphatidylcholine and cholesterol in the ratio of 97.5:2.5. It was concluded from the study that topical immunization with the developed vesosomes leads to a significant increase in IgG levels (*p* < 0.05) as compared to the conventional liposomal formulations given topically. However, the method of preparation of vesosomes at the laboratory scale is complex leading to uncertainty in its industrial scalability [37].

### 3.5. pH Fusogenic Liposomes

The pH-sensitive liposomes have evolved as a promising strategy in the intracellular delivery of drugs/antigens. Liposomes of a specific composition, typically containing either a weakly acidic or weakly basic amphiphile carry a net neutral charge and are found to be stable at physiological pH (7.4). However, following cellular internalization through receptor-mediated endocytosis, the liposomes encounter acidic conditions, and thereby undergo a phase transition owing to partial protonation of the weakly acidic groups. This in turn results in the destabilization of the membranes followed by the release of the payload into the cytoplasm. Dioleoyl phosphatidylethanolamine (DOPE) and cholesteryl hemisuccinate (CHEMS) are the most commonly used lipids that confer fusogenic properties to liposomes [38]. The selection of amphiphilic stabilizers and their molar percentage with regard to the phosphatidyl ethanolamine content determine the extent of internalization, fusogenic ability, and stability in biological fluids [39]. In certain cases, the liposomes made of pH-insensitive phospholipids tend to undergo pH-dependent fusion in the presence of toxins, viral particles, or proteins. Any variation in the pH causes conformational rearrangement in the viral proteins and subsequent destabilization of the membrane [40]. The pH-sensitive liposomes are promising vehicles for the delivery of peptides enabling their use as prophylactic or therapeutic vaccines. Watarai et al. conducted a study in which liposomes were modified using a pH-sensitive polymer with fusogenic properties, named succinylated poly(glycidol), and further tested its effectiveness as an antigen carrier. The immunization of mice with ovalbumin-containing modified liposomes subsequently resulted in the induction of a significantly higher amount of ovalbumin-specific IgG3, IgG2a, and IgG1 antibody titers in comparison to the unmodified liposomal groups [41]. Chang et al. developed pH-sensitive liposomes encapsulating V3-loop peptide, a neutralizing determinant of the human immunodeficiency virus in order to stimulate virus-specific cell-mediated and humoral immune responses. V3-loop peptide contains T-cell epitopes of the HIV glycoprotein making it an important sequence for the formulation of vaccines. The study concluded that the developed system was able to elicit the production of cytotoxic T-lymphocytes (CTL) and virus-specific neutralization antibodies whereas no response was elicited with the immunogen in the absence of liposome encapsulation [42]. Lee et al. studied the immunization potential of fluorescein isothiocyanate-conjugated H-2Kb CTL epitope encapsulated in pH-sensitive liposomes. Following three days of immunization, significant activation of CTL responses induced by the delivered antigen was observed. It can be concluded from these studies that pH-sensitive liposomes act as strong peptide adjuvants making them promising carriers for the development of prophylactic and therapeutic vaccines [43]. Poly(glycidol) derivatives such as 3-methylglutarylated poly(glycidol) (MGlu-PG) and succinylated poly(glycidol) (Suc-PG) have been studied for modification of liposomes owing to their ability to confer fusogenic properties at mildly acidic pH. The liposomes modified with poly(glycidol) derivatives possess carboxyl groups in the polymeric side chain that provides a negative charge to the liposome’s surface. Therefore, such liposomes are taken up preferentially by the dendritic cells resulting in efficient activation of CTLs [44]. Yuba et al. prepared ovalbumin-loaded pH-sensitive liposomes through surface modification of dioleoyl phosphatidylethanolamine/egg yolk phosphatidylcholine (DOPE/EYPC) lipid vesicles with MGlu-PG of linear (MGlu-LPG) and hyperbranched structure (MGlu-HPG). The study demonstrated that upon subcutaneous or nasal administration, the modified liposomes were taken up more effectively by the dendritic cells inducing stronger OVA-specific cellular immune responses in comparison to the unmodified liposomes. Further, tumor suppression in around 50–75% of the mice establishes the potential of the modified liposomes to elicit a strong cellular immunity required for the regression of OVA-expressing tumor cells (Figure 4) [45]. The ability of MGlu-HPG to form a higher number of hydrophobic domains on exposure to a weakly acidic environment as compared to MGlu-PG confers higher membrane disruption ability to MGlu-HPG. Further, the membrane fusogenic property and subsequent cellular association of MGlu-HPG increases with the degree of polymerization indicating that the bulkier polymer-modified liposomes can be efficiently recognized by the scavenger receptors on dendritic cells [46]. 

### 3.6. Ethosomes

Lipid-nanocarriers that are composed of a relatively high amount of ethanol, approximately 25 to 40%, are termed ethosomes. The presence of ethanol in lipid vesicular structures increases the permeation at the stratum corneum enabling transdermal delivery of drugs and antigens across the skin [47]. In a comparative study on the skin penetration properties of ethosomes with other liposomal derivatives tagged with ovalbumin peptides, it was found that a strong immune response was elicited upon delivery of the peptides by ethosomes owing to enhanced permeation into the deeper regions of the skin [48]. Zhang et al. formulated an ethosomal carbomer hydrogel containing 30% ethanol that was able to trigger a proper humoral immune response on applying to the skin of mice [49]. Raghuvanshi et al. developed a single-dose HBsAg ethosomal vaccine for administration through the nasal route. The developed formulation demonstrated a higher immunological response as compared to the alum-HBsAg vaccine. A single dose of ethosomal vaccine showed effective and measurable immunoglobulin and cytokine levels eliminating the need for a booster dose of vaccine. This serves as a proof of concept for the utilization of ethosomes as a potential alternative to conventional needle-based vaccination [50].

### 3.7. Transferosomes

Owing to structural resemblance with cellular vesicles/exocytotic vesicles, transferosomes (also referred to as ultra-deformable liposomes) have been widely investigated for the transport of active drug substances and antigens. The membrane structure of transferosomes comprises phospholipids, an edge activator, and a surfactant molecule with a single chain. The presence of edge activators such as sodium deoxycholate, Span (60, 65, 80), sodium cholate, dipotassium glycyrrhizinate, and Tween (20, 60, and 80) differentiates transferosomes from the conventional liposomes. The role of edge activators in transferosomes is to cause destabilization of the phospholipid bilayer that enhances the deformability of the nanocarrier by reducing its interfacial tension [51]. This ultra-deformability of transferosomes makes it ideal for transdermal delivery and the diffusion of antigens to the APCs, thereby making it superior in comparison to conventional liposomes [52]. However, different studies have reported contradictory results regarding the ability of transferosomes in enhancing skin permeability. For instance, delivery of the Hepatitis B surface antigen by means of a plasmid DNA-cationic transferosome complex demonstrated superior levels of humoral and cellular immune responses as compared to vaccination with conventional liposomes [53], whereas Ding et al. reported that transcutaneous immunization with transferosomes alone did not enhance immunogenicity but required skin pre-treatment with microneedles [54]. Wu and his co-workers designed transferosomes with positive and negative surface charges and integrated them with hyaluronic acid microneedles for transdermal immunization. It was observed that the cationic nano vaccines were more efficient in activating the maturation of dendritic cells and inducing Th1 immunity in comparison to their anionic counterparts as suggested by a significant increase in IgG2a/IgG1 ratio and elevated cytokine secretion from Th1 cells without an enhancement in the Th2 response [55]. This enhanced Th1 antigen-specific immune response in lymph nodes through a transdermal vaccine delivery system can be used as a potential approach for immunotherapy.

### 3.8. Immuno-Liposomes

Immunoliposomes also referred to as antibody-directed liposomes have been recognized as a potential tool for the site-specific delivery of drugs/diagnostic agents/antigens. The coupling of antibodies onto the surface of liposomes allows for active tissue targeting via binding to the receptors located on the cell surface [56]. Eskandari et al. prepared immunoliposomes by grafting non-immune mouse IgG onto the surface of liposomes. They further evaluated the effect of immunoliposomes on the intensity and type of generated immune response against Leishmania, and also compared their efficacy with conventional liposomes. The results of the study suggested that immunoliposomes induced strong cell-mediated immune responses against L. major challenge in BALB/c mice. Thus, it can be concluded that immunoliposomes can be used as a potential strategy for immunization against L. major [57]. In another study, LAG3 (lymphocyte activating gene 3-IgG) and P5 peptide were coupled into the surface of PEGylated liposomes for simultaneous co-delivery and were investigated for their therapeutic effectiveness in a mice model of TUBO breast cancer. A higher percentage of CD8^+^ and CD4^+^ T cells in the spleen followed by a pronounced and rapid infiltration of these effector cells into the tumor site were observed. These observations suggest that LAG3-Ig-P5-immunoliposomes can serve as a potential candidate for developing a highly targeted therapeutic cancer vaccine in the treatment of HER2/neu^+^ breast cancer [58]. A similar study was performed by Rodalec and his coworkers in which immunoliposomes made of synthetic lipids (ANC-2) and natural (antibody nanoconjugate-1 [ANC-1]) encapsulating docetaxel to treat HER2-positive breast cancers were developed. In vitro study results suggested significant antiproliferative efficacy and targeted drug delivery in breast cancer [59].

### 3.9. Lipid Nanoparticles

The recent development of lipid nanoparticle-based mRNA vaccines for combating COVID-19 has thrown a spotlight on the emerging role of these nanocarriers as potential vehicles for the delivery of a wide variety of therapeutics including antigens. Owing to some of the limitations associated with the early generation of lipid-based carriers, i.e., the liposomes, the next generation of lipid nanoparticles (LNPs) have emerged as frontiers in drug and vaccine delivery. LNPs mainly include solid–lipid nanoparticles (SLNs) and nanostructured lipid carriers (NLCs) [60]. 

SLNs are composed of solid lipids whereas NLCs are composed of mixtures of solid and liquid-crystalline lipids. In comparison to conventional liposomes, SLNs, and NLCs have considerably high loading capacities. Additionally, the reduced mobility of molecules in the solid state allows these carriers to control the release of their drug/antigen payloads more effectively. Structurally, these carriers are composed of lipids and stabilizing agents such as surfactants along with other coating materials. Typically, LNPs comprise four lipid components, namely, ionizable lipid, cholesterol, phospholipids, and PEGylated lipid [61]. Ionizable lipids are most commonly used in the preparation of LNPs owing to their nontoxicity in comparison to non-ionizable lipids. The ionizable lipids remain neutral at physiological pH and upon cellular uptake, they become positively charged owing to the acidic pH of the endosomes. Following protonation, they interact with the anionic membrane of the endosome to form ion pairs thereby facilitating membrane fusion, endosomal escape, and subsequent release of the cargo into the cytosol (Figure 5) [62]. 

SLNs comprising cationic lipids can bind to negatively charged DNA molecules resulting in an SLN/DNA complex that has the potential to be used as a vaccine. Various studies have been conducted to explore the potential of SLNs as vaccine carriers [63]. Francis et al. developed an adjuvanted solid lipid nanoparticle (SLN-A) for use as a DNA vaccine carrier encoding the Urease alpha (UreA) antigen from Helicobacter pylori. SLNs were synthesized by a modified solvent emulsification technique and were composed of cationic lipids monophosphoryl lipid A as an adjuvant. The developed formulation was evaluated for zeta potential, morphology, and in vitro transfection capacity. In vitro study results showed that the developed formulation can be efficiently transfected in murine immune cells for expression of recombinant Helicobacter pylori antigen Urease A, thus demonstrating their potential for vaccine delivery [64]. A comparative study was performed to evaluate two vaccine delivery strategies namely electroporation and a cationic SLN formulation in the administration of a DNA vaccine harboring Leishmania donovani A2 antigen along with Leishmania infantum cysteine proteinases. Further, its potential against Leishmania infantum challenge was evaluated. The study results demonstrated that similar to the electroporation delivery system, SLNs as a nanoscale vehicle of Leishmania antigens could enhance immune response, hence indicating the effectiveness of these approaches against visceral leishmaniasis [65].

Gerhardt and his colleagues developed thermostable and lyophilizable NLCs for the delivery of replicating RNA-based vaccines administered through the intramuscular route. The developed NLCs were composed of liquid lipids (squalene) and solid lipids (trimyristin) surrounded by surfactants (sorbitan monostearate and polysorbate 80) and a cationic lipid (1,2-dioleoyl-3-trimethylammonium-propane) [DOTAP]. The system was complexed with mRNA and was able to retain stability at room temperature for around 8 months and at refrigerated temperature for around 21 months [66]. A live-attenuated RNA hybrid vaccine composed of an in vitro transcribed and highly attenuated CHIKV genome encapsulated in a stable NLC formulation has been developed by Voigt and his co-workers [67]. In another study, Kaur et al. prepared NLC from didodecyldimethylammonium bromide (DDAB), oleic acid and co-encapsulated a Pam 2 CS derivative (T-2, TLR2 agonist) with an imidazoquinoline derivative (T-7, TLR7 agonist) as a combination vaccine adjuvant. The developed formulation exhibited high TLR7 and TLR2 agonistic activity. The adjuvant potency of this system was also evaluated in mice with influenza hemagglutinin protein and recombinant hepatitis B surface antigen. The study results demonstrated that NLCs containing T-2 and T-7 induced good efficacy in mice challenged with a lethal dose of influenza virus [68].

## 4. Factors Influencing the Efficacy of Lipid-Based Adjuvants for Vaccine Delivery

The lipid composition of the nanocarrier and its bio-physical formulation characteristics have a significant impact on the adjuvanticity of vaccines. These characteristics include lamellarity, the surface charge of the membrane, vesicular size, fluidity of the bilayer membrane, and the presence of lipids that are immunostimulatory [69].

### 4.1. Surface Charge

The presence of charge on a nanocarrier’s surface has a substantial impact on its ability to stimulate the immune system. It plays a crucial role in influencing antigen loading, its subsequent release, and in maintaining the stability of the vesicles. Because of surface charge, electrostatic interactions of the nanocarriers with various biological components, immune cells, and even intracellular organelles tend to occur [70].

Charged lipid-based delivery systems are usually prepared by incorporating negatively and positively charged phospholipids. The negatively charged phospholipids such as 1,2-dioleoyl-sn-glycero-3-phosphoethanolamine and 1,2-dioleoyl-sn-glycero-3-[phospho-rac-(1-glycerol)] (DOPG) are generally used for the preparation of anionic vesicles [71]. Similarly, phospholipids such as N-[1-(2,3-dioleoyloxy) propyl]-N,N,N-trimethyl-ammonium methyl sulfate, 1,2-dioleoyl-3-trimethylammonium-propane (DOTAP), oleic acid, dioleoylphosphatidyl ethanolamine (DOPE) and dimethyldioctadecylammonium bromide induce a positive charge on the surface leading to the formation of cationic lipid-vesicles [72].

Cationic lipids are considered superior as a vaccine adjuvant in comparison to anionic and zwitterionic (neutral) lipids as they can interact with negatively charged antigens. The resulting electrostatic interactions in turn will prevent the immediate release of antigen following administration leading to the formation of a depot to recruit APCs (Figure 6). APCs, being negatively charged can adhere to cationic vesicles which thereby results in accelerated cellular phagocytosis. This eventually leads to an enhanced contribution to APC activation through the intracellular FcRγ–Syk–Card9 pathway [73,74]. Immunostimulatory activity of cationic lipids has also been reported. However, studies have demonstrated that when liposomes composed of cationic lipids were injected intravenously into mice, hemolytic as well as cell-disrupting activity is observed which restricts their application for immunization purposes. This activity may be attributed due to the presence of a quaternary ammonium group in cationic lipids [75]. The surface charge density of lipid-based carriers also plays a vital role as it has to be above a certain threshold for its utility as a vaccine adjuvant. A study conducted by Ma et al. showcased the impact of surface charge density on the immune responses mediated by DOTAP/DOPC cationic liposomes. The liposomes with a higher charge density such as DOTAP/DOPC 5:0 and 4:1 liposomes were found to significantly enhance the maturation of dendritic cells, antigen uptake, ROS generation, and the secretion of OVA-specific IFN-γ and IgG2a. In contrast, liposomes with a low-charge density such as DOTAP/DOPC 1:4 liposomes were not able to stimulate immune responses even at significantly higher concentrations thus establishing the dependence of the immunoregulatory effect of cationic lipid carriers on its surface charge density [76].

### 4.2. Route of Administration

Various routes have been explored by researchers for administering lipid-based drug delivery systems as a vaccine adjuvant. They can be delivered mainly through the following four ways: oral uptake, injectional route (intramuscular, intravenous, subcutaneous, intradermal, and intraperitoneal injection), inhalational and topical administration (e.g., mucosal and cutaneous administration) [77]. Schmidt et al. performed a study to evaluate the impact of the route of administration of a CAF09-adjuvanted liposome on the induction of CD8^+^ T-cell response. Fast drainage of CD8^+^ cells to the local lymph nodes occurred on intraperitoneal immunization whereas immunization through the intramuscular route resulted in the occurrence of a depot at the site of injection. This in turn prevented self-drainage of the vaccine to the lymph nodes and the spleen at a level that is sufficient enough to pass the threshold for the activation of CD8^+^ T-cells. It was observed that immunization through the intraperitoneal route facilitates the concomitant delivery of the CAF09 adjuvant and the OVA antigen to cross-presenting CD8α^+^ dendritic cells located in the lymphoid tissues. Furthermore, CAF09 led to the stimulation of dendritic cells within the first 24 h after intraperitoneal but not intramuscular immunization [78] (Figure 7).

### 4.3. Membrane Fluidity

The membrane flexibility/fluidity of lipid vesicles has been demonstrated to have a crucial impact on the activity of the adjuvant effect of lipid-based nanocarriers [79]. Membrane fluidity is dependent on the saturation as well as the length of lipids used for the preparation of lipid carriers. The bilayer’s physical state (i.e., the gel state or liquid-crystalline state) has been found to affect the cellular uptake, intracellular trafficking, and the subsequent processing of the vaccine components that in turn may impact the immunological responses. The lipids undergo certain changes upon reaching the phase transition temperature (T_m_). The reduction in T_m_ can be achieved either by reducing the length of the hydrocarbon chain or through the insertion of carbon double bonds [13]. These modifications can be utilized to increase the immunostimulatory activity of lipid-based vaccine delivery systems.

Various studies have suggested that lipid vesicles made of “fluid” phospholipids, characterized by a low transition temperature are taken up by cells primarily as a result of fusion with the plasma membrane whereas lipid carriers that are “solid” (composed of phospholipids with a high transition temperature) enter into the cells specifically via an endocytic process [80]. Yasuda and his colleagues studied the ability of liposomes composed of phosphatidylcholine with varying transition temperatures, i.e., DLPC: −2 °C; DMPC: 23 °C; DOPC: −19 °C and DSPC: 54 °C to induce the production of antibodies against a lipid-linked dinitrophenyl hapten (DNP-Cap-PE). This was measured through a plaque assay involving the quantification of spleen antibody-secreting cells. Cholesterol was added to each formulation. It was observed from the study that lipids with a higher transition temperature demonstrated higher anti-DNP antibody response over a 4-fold range and a correlation was also observed between the transition temperature of lipids and the antibody response for DMPC, DLPC, DSPC, and DOPC [69]. The incorporation of cholesterol in the liposome is known to modulate membrane fluidity and is therefore widely utilized to improve the stability of liposomes. Some studies have also demonstrated the immunostimulatory effect of cholesterol when incorporated into liposomes. Van et al. demonstrated an increase in the humoral immunogenicity of liposomes when increasing the cholesterol amount in the membrane of liposomes [81].

### 4.4. Vesicle Size

Numerous investigations have demonstrated the effect of average vesicle size on the induction of the type of immune response following immunization. It has been also reported that vesicle size governs whether an immune response will be proceeded by Th1- or Th2-based pathways. Brewer et al. in his studies showed that inoculation of lipid carriers possessing an average size of ≤225 nm induced Th1 responses characterized by increased IFN-γ production by the cells of the lymph node and enhanced titers of IgG2a in plasma whereas when a similar formulation with an average size of ≤155 nm was inoculated, a Th2 response was observed which can be identified through titers of IgG1 in the absence of IgG2a secretion and enhanced lymph node IL-5 production [82]. Lacey et al. explored the effect of vesicle size of the cationic liposome CAF01 comprising dimethyldioctadecylammonium (DDA) and trehalose dibehenate (TDB) on the in vitro cellular uptake and in vivo pharmacokinetics. Though no differences were observed in the draining of the vaccine from the site of injection, significant differences in the liposomal movement to the popliteal lymph node were observed [13]. Kaur and his co-workers demonstrated the combined effect of the manipulation of surface pegylation and reduction in the vesicular size of cationic liposomal adjuvants on their biodistribution and the rate and type of T cell response. The results from biodistribution studies demonstrated that the highly pegylated small unilamellar vesicles containing around 25 mol% PEG enhanced drainage and clearance to the local lymph node with a concomitant increase in the levels of antigen at the draining lymph node [83].

## 5. General Fabrication Methods of Various Lipid-Based Vaccine Delivery Systems

### 5.1. Thin Film Hydration Method

The thin film hydration method is one of the most widely used methods for the preparation of lipid-based vaccine delivery systems such as liposomes, solid-lipid nanoparticles, bilosomes, transferosomes, and ethosomes. In this method, the lipid components are dissolved in a suitable organic solvent to ensure the production of a homogenous mixture of lipids. Once the lipid mixture has been extensively mixed with the organic solvent, the organic solvent has to be evaporated to yield a dry and thin lipid film. Small quantities of organic solvent (below 1 mL) can be evaporated in a fume hood using an argon or dry nitrogen stream and for the larger volume of organic solvents, a round-bottomed flask should be used. Further, the dried lipid film is hydrated by adding an aqueous medium to it. The lipids expand and separate from the round bottom flask’s wall after being hydrated. The hydration medium’s temperature needs to be higher than the lipid’s gel-liquid crystal transition temperature (T_c_ or T_m_). The lipid suspension should be kept above the T_c_ during the hydration interval after the hydrating medium has been added. Although hydration times may vary slightly depending on the kind of lipid and its structure, it is strongly advised to hydrate lipids for an hour with rapid shaking, mixing, or stirring. The intended application of the lipid vesicles typically determines the hydration medium. Distilled water, buffer solutions, saline, and non-electrolytes such as sugar solutions are all acceptable hydration mediums. The commonly used mixtures of 0.9% saline, 5% dextrose, and 10% sugar satisfy these requirements [84]. The hydration step causes the lipid to swell and hydrate which results in the development of a highly heterogeneous MLV suspension in terms of lamellarity and size. Therefore, to attain uniformity in the size of vesicles, the final stage of the manufacturing operation involves decreasing the lamellarity and size of the lipid vesicles. In general, lipophilic drugs need to be dissolved in the phospholipid mixture before the formation of the thin film, whereas hydrophilic drugs can be introduced within the hydration medium and then can be passively absorbed into the lipid vesicles during the hydration process. However, in the case of vaccine delivery, the antigens are either dissolved in the aqueous phase followed by subsequent encapsulation in the condensation phase or are attached/complexed/adsorbed to the final vesicles in a second step. For example, in order to achieve transcutaneous immunization of ovalbumin, Li et al. prepared liposomes using the thin-film hydration method. Phospholipid S 100 and cholesterol were dissolved in 15 mL of chloroform and dried to form a thin lipid film. 25 mg of the antigen was dissolved in distilled water and the resulting aqueous phase was used to hydrate the lipid film. Hydration of the lipid film resulted in the formation of vesicular structures entrapping the antigen in the aqueous core [85]. The film hydration method for liposome preparation is depicted in Figure 8.

Barnier-Quer et al. investigated the effect of the antigen-loading method, i.e., adsorption versus encapsulation on the immunogenicity of influenza-haemagglutinin-loaded liposomes. Liposomes composed of DC-Chol: DPPC in the molar ratio of 1:1 were prepared using the thin film hydration method. It was found that the particle size, zeta potential, and loading efficiency were similar in both cases of antigen loading. However, upon subcutaneous administration, it was observed that the antigen adsorbed onto the liposomes exhibited higher antibody titers as compared to the encapsulated liposomes [86]. Conacher et al. used the thin film hydration method for the preparation of bilosomes containing the influenza subunit vaccine. The lipid constituents 1-monopalmitoyl glycerol, cholesterol, and dicetyl phosphate in the molar ratio of 5:4:1 was dissolved in chloroform (10 mL) and placed in a round-bottomed flask. At 40 °C and with reduced pressure, the chloroform was evaporated to form a thin film. The antigen dispersed in carbonate buffer containing 100 mg of bile salts was added followed by gentle swirling for 15 to 20 min at 60 °C. The non-entrapped antigens and bile salts were removed by centrifugation at 100,000× *g* for 45 min at 4 °C. The prepared vesicles were found to be stable and retained around 90% of the antigen upon exposure to bile salt concentrations that mimic normal body levels [87]. In a study by Wu et al. antigen-loaded transferosomes with opposite charges were prepared by the thin film hydration method and integrated with dissolving microneedles to elicit an enhanced Th1 antigen-specific immune response in lymph nodes. To prepare anionic transferosomes, the anionic surfactant sodium cholate was added to the antigen-containing aqueous phase whereas in the other case, to confer a positive charge to the transferosomes, stearyl amine was incorporated in the organic phase. Further, the cationic surfactant polyquaternium-7 (PQ-7) was added to the hydration process to stabilize the transferosomes [55]. Hence, it is evident from the above examples that the thin film hydration method is one of the most versatile techniques as it allows the incorporation of a wide range of materials irrespective of the solubility for the stabilization of the prepared vesicles. However, the principal limitations of Bhangham’s method are its small-scale manufacturing, difficulties in removing the organic solvent, and low entrapment efficiency [88].

### 5.2. Microfluidization

In the microfluidic approach, lipids that have been dissolved in the organic solvent such as ethanol or isopropanol are sequentially injected into microchannels that have a cross-sectional area of 5–500 µm. The fluids are pumped through an aperture by a microfluidizer at a very high pressure (10,000 psi, 600–700 bar). Focused between two water streams in a microfluidic channel (microchannel), the alcoholic phospholipid solution produces a hydrodynamic laminar flow and a diffusive mixing at the two (liquid) interfaces that encourage the lipids to self-assemble into vesicles. This technique enables the creation of tiny (monodisperse) lipid-based nanoformulations with regulated sizes and distributions using low-toxicity solvents by carefully controlling the mixing and fluid flow rates (such as ethanol) [89,90]. In recent times, the microfluidics approach has gained wide attention for the production of antigen-loaded lipid-based carriers as it offers numerous advantages such as high throughput optimization of the developed formulation, high reproducibility, precise size controllability, and a continuous production process. These features of the microfluidics device enable easy scalability and transition from laboratory scale-use to practical applications. The microfluidic device has been explored for the production of mRNA-based lipid nanoparticles, where the ethanol-dilution method is mostly used. In this method, the lipids are dissolved in ethanol and the mRNA is dissolved in suitable buffer solutions such as citrate, acetate, or malic acid buffer. Further, upon injection of the solutions into a microfluidic device, mRNA-lipid complexes are formed through electrostatic interactions that in turn reassemble to form lipid nanoparticles. Fluid dynamics play a significant role in the control of particle size and size distribution. It has been observed that when ethanol is rapidly diluted with a buffer solution, small-sized nanoparticles are formed whereas large-sized nanoparticles occur under slow dilution conditions [91]. In a study by Elia et al. lipid nanoparticles encapsulating the Fc-conjugated receptor-binding domain (RBD-hFc) of SARS-CoV-2 were prepared by a microfluidic mixer device. A mixture of lipids containing ionizable lipid, distearoylphosphatidylcholine, cholesterol, and DMG-PEG (myristoyl diglyceride) in the molar ratio of 40:10.5:47.5:2 was mixed with three times the volume of mRNA in acetate buffer and injected into the device at a flow rate of 12 mL/min. The resulting mixture was dialyzed against phosphate-buffered saline in order to remove ethanol. The mRNA-LNPs produced by this method were found to possess small and uniform hydrodynamic diameters thereby establishing the ability of this technique to achieve precise size controllability [92].

### 5.3. Detergent Depletion Method

This approach allows phospholipids to come into close contact with the aqueous phase through the utilization of detergents which associate with the phospholipid molecules and serve to screen the hydrophobic region of the molecule from water. As a result, mixed (detergent/lipids) micelles are created. High CMC detergents like Triton X-100, sodium deoxycholate, sodium cholate, and alkyl glycoside are the frequently used cleaning agents [93]. The dilution approach using a buffer (10- to 100-fold) is the easiest way for removing detergents. The size and polydispersity of the original micelles grow after the aqueous solution of a mixed lipid-detergent system is diluted with a buffer. Finally, once the solution is diluted over the boundary of the mixed micellar phase, a spontaneous shift from polydispersed (elongated) micelles to vesicles takes place. In a study, the aggregates in the lecithin-bile salt (detergent) mixed system’s aqueous solution evolved from spherical micelles to longer (flexible) cylindrical micelles to almost monodisperse unilamellar vesicles when the dilution factor was increased (at the higher dilution factors). The principle of spontaneous curvature can be used as the theoretical basis for an explanation of this event (and the critical packing factor). In brief, (proteo) lipid vesicles will develop during the last step of the detergent removal procedure when the overall detergent concentration falls below the CMC of the detergent, and further techniques should be utilized to remove any leftover detergent that is present in the nanoformulation. Some peptides, proteins, and oligonucleotides are sensitive to organic solvents and may undergo denaturation upon exposure to such solvents. The detergent depletion method may be a potentially scalable technique for such agents. mRNA-based liposomes encoding the influenza virus nucleoprotein were prepared by the detergent removal technique for the induction of virus-specific cytotoxic T-lymphocytes. Initially, the lipid mixture containing 30 µmoles of cholesterol-dipalmitoyl phosphatidylcholine-phosphatidylserine was dried by rotary evaporation and subsequently resolubilized with 300 µmoles of n-octyl-beta-D-glucopyranoside (a non-ionic detergent) in 1 mL of water. To the micellar preparation obtained, 50 µg of the nucleoprotein was added and subjected to dialysis for 16 h. The dialysate was extruded through a polycarbonate membrane with a pore size of around 200 nm to obtain a homogenous suspension of liposomes. Further, the unencapsulated mRNA was removed by passing the suspension through a gel-permeation column [94]. Virosomes are commonly prepared by the detergent removal/solubilization method. For example, in the preparation of influenza virosomal vaccines, the influenza virus is initially solubilized and isolated by detergents. The virus subunits are then concentrated using a density-gradient centrifugation-vacuum dialysis method and incorporated into prepared liposomes. During this process, the subunits get randomly integrated into the lipidic membrane either into the cavity or on the surface. In the case of the influenza virus, glycoproteins are inserted into the lipid bilayer by means of a hydrophobic effect and get stabilized by van der Waals forces [23]. The primary limitations of the detergent removal approach include challenges related to the removal of residual amounts of detergents, attainment of a final low concentration of lipid vesicles, and low hydrophobic chemical entrapment effectiveness. Further, the multi-step process makes it time consuming and costly for industrial-scale applications [88].

### 5.4. Ethanol Injection Method

The use of the ethanol injection method results in the formation of lipid vesicles ranging between 30–170 nm and is widely used to prepare SUVs. The lipid content and injection speed have a direct effect on the size. This technique involves injecting lipids that have been dissolved in an organic solvent (in this example, ethanol) into the water phase while stirring followed by the removal of solvent to create lipid vesicles. The mixture is then allowed to hydrate while being stirred for a further 15 min. Either centrifugation through a silica gel column or rotary evaporation can be used to remove the ethanol from the solution. This approach has several drawbacks including very poor hydrophilic molecule encapsulation effectiveness and low lipid concentrations in the final solution because of the high ethanol content. Additionally, ethanol concentration should not be higher than 7.5 percent in order to prevent destabilization of the vesicles which reduces the permissible amount of lipids. However, apart from these, the technique is highly beneficial for producing industrial amounts of liposomes [84]. Solid lipid nanoparticles for the delivery of Hepatitis B surface antigen were prepared by the solvent injection method in which the organic phase containing 50 mg of tristearin in acetone was injected rapidly through a needle into an aqueous phase containing lactose monohydrate, antigen, and Tween-80. The critical parameters that influenced the characteristics of the final dispersion were lipid concentration, surfactant concentration, stirring speed, and stirring time. The optimized formulation exhibited an entrapment efficiency of around 65% [95].

### 5.5. Reverse Phase Evaporation

The formation of inverted micelles is the fundamental principle of reverse-phase evaporation. Lipids are dissolved in an organic solvent using this approach, which encourages the development of inverted micelles. After that, a predetermined amount of an aqueous phase (buffer) is added to the solution. A water-in-oil (W/O) microemulsion is produced when the lipids rearrange themselves at the water-oil interface. To aid in the creation of a homogenous dispersion, the W/O microemulsion can be emulsified using mechanical or sonication techniques. The organic solvent can be eliminated by using continuous rotating evaporation (under decreased pressure) up until the production of a thick gel. The breakdown of inverted micelles and the subsequent production of lipid vesicles are both favored by slower organic solvent removal (LUVs) [88]. The aqueous volume to lipid ratios of reverse phase evaporation lipid vesicles are four times greater than those of hand-shaken lipid vesicles or multilamellar lipid vesicles and they may be formed from a variety of lipid formulations [96]. Hepatitis B antigen-loaded solid lipid nanoparticles modified with TLR-4 agonist were prepared by the reverse phase evaporation method for effective colonic uptake of the developed vaccine [97]. Though the reverse phase evaporation method is suitable for the encapsulation of large amounts of macromolecules, it is seldom used in the manufacturing of biomolecule-loaded lipid carriers including peptides, oligonucleotides, and enzymes. Thus, this technique has not been explored widely for vaccine delivery systems. The presence of residual solvents and the difficulties associated with its scale-up further limits its use as a vaccine delivery carrier [98].

### 5.6. Other Methods

#### 5.6.1. Freeze Drying Method

Water-soluble components incorporated in lipid-based nanoformulations are prone to leakage whilst preparation, storage, and shelf life. So, these problems can be solved by using the freeze-drying method. The freeze-drying technique is an ideal solution to the long-term stability issues of heat-sensitive thermolabile products. This method is based on the removal of water from lipid-based products in the frozen state at extremely low pressures in the presence of certain sugars (sucrose, trehalose) to prevent the leakage of encapsulated material [93]. In a study by Gao et al., the model antigen ovalbumin was encapsulated into chitosan-modified squalene nanostructured lipid carriers for exploring its potential in activating immune responses. The formulation upon subjection to freeze-drying exhibited good stability. To avoid any change in the physicochemical properties of the nano-carrier after redispersion, 10% *w*/*v* of sucrose was incorporated as the lyoprotectant. The absence of lyoprotectants makes the formulation unstable. This was evident upon the redispersion of the ovalbumin-loaded nanocarriers that were lyophilized without a lyoprotectant. An increase in the particle size and PDI was observed and a change in zeta potential from positive charge to negative charge occurred thereby explaining the significance of the incorporation of an optimum concentration of lyoprotectant during the freeze-drying process [99]. Thin Film Freeze-Drying (TFFD) is an ultra-rapid method of freezing vaccines that involves placing a solution or suspension of the antigen on a cryogenic surface that is pre-cooled. Post-impact, the droplets spread into a thin film which is rapidly frozen and dried to yield powders that exhibit excellent aerosol performance [100]. Dry powder formulations of AS01B-adjuvanted ovalbumin vaccines were prepared using the TFFD method to study the integrity of the formulation and aerosol performance characteristics. AS01B is a liposomal formulation composed of two immunostimulants namely 3-O-desacyl-4-monophosphoryl lipid A and QS-21. It was found that upon subjecting the AS01B liposomal adjuvant to TFFD using 4% *w*/*v* sucrose as the stabilizer, no changes in the particle size distribution occurred after the reconstitution step. Further, desirable aerosol properties for efficient pulmonary administration were achieved including a mass median aerodynamic diameter of 2.4 ± 0.1 µm and a fine particle fraction of 66.3 ± 4.9% [101].

#### 5.6.2. Supercritical Fluid-Assisted Method

Conventional methods of preparation of lipid-based nanocarriers using organic solvents, may degrade the encapsulated drug/antigen and also cause toxicity to human health and the environment. The preparation of lipid nanocarriers by supercritical fluid technology overcomes these problems. SC-CO_2_ is used in supercritical fluid technology. This fluid is maintained at supercritical temperature and pressure. In this state, the fluid efficiently works as a solvent for phospholipids. As a result, the SCF can easily replace organic solvents in the preparation of lipid nanocarriers. The phospholipids are first dissolved in SCF and then a co-solvent mixture with the aqueous phase is decompressed by spraying it (via a nozzle). When a fine droplet of a co-solvent mixture comes into contact with a high-pressure environment, the solvent evaporates resulting in the formation of organic solvent-free lipid carriers [88]. A comparative study on the encapsulation efficiencies of bovine serum albumin (BSA)-loaded liposomes prepared using a supercritical fluid-assisted process and those prepared using the conventional thin-film hydration method was carried out by Campardelli and co-workers. Very high encapsulation efficiencies of around 92–98% were obtained with the supercritical fluid-assisted process whereas, in the case of the conventional method, only an encapsulation efficiency of 57% was achieved. This study paves way for the potential use of the SCF-assisted method in the preparation of antigen-loaded nanocarriers owing to its ability to confer a high-antigen loading capacity [102].

## 6. Liposomes as Adjuvants

### 6.1. Liposomal DNA as Adjuvant

Adjuvants are compounds that enhance the immune response of antigens by exposing them to the pattern recognition receptors of APCs. Liposomes act as delivery carriers for antigens and immunopotentiators and thereby are highly versatile vaccine adjuvants. The adjuvant mechanism of liposomes is based on their ability to interact with APCs which subsequently enhances the delivery of immunopotentiators and antigens to APCs [103]. Liposomes can be tailored to achieve the desired immune response depending on the choice of lipid composition, physicochemical characteristics such as particle size, surface charge, membrane fluidity, and hydrophobicity, the method of incorporation or loading of the immunostimulators and their mode of action [104]. CpG oligonucleotides encapsulated in liposomes are well known to be potent immunoadjuvants for a wide array of antigens. These are composed of unmethylated CpG dinucleotide sequences and are present at a higher concentration in bacterial DNA. The CpG motifs are recognized by the Toll-like receptors located on the B-lymphocytes and dendritic cells that in turn trigger an immune cascade. Therefore, the co-encapsulation of synthetic CpG motifs along with antigens in liposomes act as potent adjuvants by enhancing the uptake of antigens, their presentation by the APCs, and the secretion of chemokines, cytokines, and antibodies thus enhancing the immune response making liposomal DNA a promising adjuvant for the delivery of vaccines [105]. Comussone et al. explored the effect of a subunit vaccine comprising four recombinant protein molecules of S. aureus namely α-toxins, β-toxins, ClfA, and FnBPA against intramammary infections by co-encapsulating it with CpG oligonucleotides in liposomes. The levels of specific antibodies against the incorporated proteins namely IgG, IgG1, and IgG2 were found to be significantly enhanced in the vaccinated group [106]. Liposomes coated with mannose and the immune adjuvant CpG oligonucleotides were studied for their effectiveness in producing antitumor-specific immune responses. The liposomes were loaded with the melanoma-specific peptide TRP2180-188 and were found to be effectively taking up the dendritic cells leading to its activation and subsequent upregulation of MHC II, CD80, and CD86. Further, the liposomes were found to effectively reduce the growth of the implanted melanoma by suppressing tumor angiogenesis. Thus, TRP2-CpG-liposomes can be attributed as a novel vaccine formulation that is capable of enhancing the antitumor responses via alleviation of the immunosuppressive environment in tumors [107].

### 6.2. Cationic Liposomes as Vaccine Adjuvants

Cationic liposomes offer a wide range of applicability as cell transfection reagents and vaccine adjuvants as the high density of positive charges enhances its tendency to get adsorbed onto the surface of negatively charged cell membranes. These liposomes enter the cell through various pathways such as clathrin-mediated endocytosis thereby mediating successful intracellular delivery of the antigen (Figure 9). Further, cationic liposomes prevent the clearance of antigens from the body causing prolonged retention of the antigen at the site of injection. This further leads to the effective presentation of the antigens to the antigen-presenting cells thereby enhancing the host’s immune response [12].

Cationic liposomes have been extensively explored as adjuvants to elicit immune responses against *Mycobacterium tuberculosis* subunit vaccines. Liposomes composed of DDA are promising TB adjuvants owing to their strong antigen absorption characteristics and the ability to produce a depot effect at the injection site. DDA in combination with monophosphoryl lipid A adjuvant has been found to induce both humoral and cellular immune responses thereby causing a reduction in the bacterial load in the lungs. The protection conferred by the liposome-adjuvant system was found to be higher than when compared to DDA alone [108]. In a recent study by Yuba et al., the adjuvanticity of polysaccharide-derived modified liposome vaccines was investigated. The incorporation of cationic lipids on β 1,3-glucan derived carboxylated polysaccharide-modified liposomes enhanced the intracellular uptake by the APCs. Further, inflammatory cytokine production from dendritic cells was also enhanced thereby establishing the effectiveness of lipid-based nano vaccines in the induction of specific immune responses [109]. A cationic adjuvant (CAF01) composed of DDA as the lipid component, the synthetic mycobacterial cord factor, α,α′-trehalose 6,6′-dibeheneate (TDB) as the immunomodulator, and a model antigen has demonstrated a strong and complex immune response both humoral and cell-mediated at significantly higher levels than the conventionally used adjuvants such as aluminum in three animal models that possess a varying immunological requirement: *Mycobacterium tuberculosis*, *Chlamydia trachomatis,* and malaria. The model antigen selected was a leading vaccine candidate with a proven record of immunogenicity markedly *Plasmodium yoelii* MSP1-19 antigen, Ag85B-ESAT-6 fusion antigen of *Mycobacterium tuberculosis* and the chlamydia vaccine antigen MOMP. The findings from the study depicted that the Ag85B-ESAT-6 fusion antigen induced a strong immune response with the IgG1 antibodies titer reaching the same levels as obtained with the Al(OH)_3_ adjuvanted vaccine. Similarly, extremely high levels of MSP1-19-specific IgG1 and IgG2a antibodies were induced by the CAF01 adjuvant, being 700 folds (IgG2a) and 10 folds (IgG1) higher than the Al (OH)_3_-adjuvanted vaccine. Further, mice immunized with the chlamydia antigen MOMP also exhibited partial protection and a decrease in the bacterial load at all-time points post-infection as compared to the mice receiving MOMP in Al(OH)_3_ [110].

Cationic lipid-DNA complexes (CLDCs) are the liposomes complexed with selected plasmid DNA that has been widely investigated for their potential as adjuvants. The complex named JVRS-100 is available as a lyophilized powder and is composed of 1-[2-(oleoyloxy)-ethyl]-2-oleyl-3-(2-hydroxyethyl imidazolinium chloride and cholesterol. CLDCs cause induction of APC uptake followed by activation of TLRs and interferons thereby causing stimulation of the adaptive immune response [5]. In a study by Bernstein et al., the use of CLDC as an adjuvant for the Herpes Simplex Type 2 Glycoprotein D vaccine demonstrated a significant reduction in the recurrence of virus shedding in a genital herpes guinea pig model which is a crucial factor for decreasing the spread of HSV-2 [111]. CLDCs have demonstrated an enhancement in the immunogenicity and cross-protective efficacy of an influenza-A H5N1 split vaccine in mice. A remarkably higher induction of virus-specific antibodies was achieved upon immunization with CLDCs when compared to the adjuvant-free formulation showing a 30-fold increase in the dose-sparing effect. The mice that received a single dose of adjuvanted CLDCs and subsequent H5N1 viral challenges exhibited mild illness, lower viral titers in the lungs, and undetectable viral titers in the spleen and brain with a 100% survival rate following the viral challenges. Further, the increase in viral titers, weight loss, and mortality in adjuvant-free vaccine recipients establish the promising potential of CLDCs as adjuvants in eliciting humoral and T-cell responses [112].

## 7. Lipid Nanoparticle-Based Messenger RNA (mRNA) Vaccines

Therapeutic applications for mRNA include virus vaccination, protein replacement therapy, cancer immunotherapy, cellular reprogramming, and genome editing. For therapeutic effects to take place, it is necessary for mRNA molecules to reach the cells of interest and to produce a sufficient therapeutic protein. However, mRNA delivery systems still face difficulties with targeted delivery and endosomal escape highlighting the need for reliable and efficient mRNA delivery platforms. The most effective method for preventing and eventually eliminating an epidemic is through the use of vaccines. Numerous materials, such as lipids, lipid-like substances, polymers, and protein derivatives have been produced for the delivery of messenger RNA [113,114,115,116]. Figure 10 represents the lipid nanoparticle encapsulated mRNA vaccines.

The first mRNA vaccination was developed in 1993 and had liposomes and mRNA expressing a nucleoprotein from the influenza virus. This vaccine was first given to mice and showed virus-specific cytotoxic T-cell responses [88]. Since then, lipid nanoparticle-based mRNA formulations have proven to be an effective replacement for older vaccine delivery systems [117,118]. Liposomes have been extensively researched and reached clinical practice for the delivery of small molecules, siRNA, and mRNA [119]. mRNA vaccines can incorporate multiple antigen mRNAs into a single vaccine. For example, a cytomegalovirus (CMV) vaccine has six mRNAs, five of which express a pentameric antigen and one of which encodes a glycoprotein antigen [120]. Unlike other vaccination platforms, such as recombinant proteins and inactivated vaccines, GMP-grade lipid nanoparticle mRNA vaccines can be manufactured for specific antigens in a relatively short amount of time. Nonetheless, one must take into account the fact that mRNA is unstable and has a relatively limited half-life. In addition, lipid nanoparticles need to be tested for safety, and it is unclear how they should be stored before they can be used in humans. Engineered mRNA molecules and lipid nanoparticles with optimized designs are being used to combat these issues. These factors have encouraged the rapid development and clinical assessment of mRNA-1273 (NCT04470427) [7] and BNT162b2 (NCT04368728) [121] COVID-19 mRNA vaccines. For both vaccines, ionizable lipid nanoparticles were used to deliver nucleoside-modified messenger RNA expressing the full-length SRAS-CoV-2 spike protein. mRNA vaccines encapsulated in lipid-based carriers are also under study as a potential influenza vaccination [122,123,124]. Phase I clinical investigations have been completed on both the mRNA-1440 and mRNA-1851 vaccines. These liposomal vaccines use mRNA encoding haemagglutinin from the H10N8 and H7N9 influenza viruses, respectively. Table 1 enlists the various clinical trials of lipid-based mRNA vaccines.

## 8. Lipid-Based Deoxyribonucleic Acid (DNA) Vaccines

DNA vaccines in contrast to conventional vaccines are bacterial plasmids developed to carry a particular encoding gene that is in charge of enabling the host to express the target antigen and trigger an immune response [126,127]. High specificity and the ability to add more sequences to the plasmid (like adjuvants that can increase the immunostimulatory effect of the expressed antigen) are two major advantages of DNA vaccines over conventional vaccines [128]. There is no requirement for live vectors or sophisticated biochemical production techniques. Many animal trials have shown promising results. Further, less manufacturing expenses, simple transport, and storage methods have increased interest in DNA vaccines. The approved uses for DNA vaccines in veterinary medicine include treating West Nile virus in horses, canine melanoma [129], infectious hematopoietic necrosis in farm-raised Atlantic salmon [130], and growth hormone-releasing hormone deficiency in pigs using gene therapy [131]. Rodriguez et al. tested the ability of plasmid DNA-encapsulated liposomes encoding *Babesia bovis* merozoite surface antigen-2c to elicit long-lasting humoral immune responses in mice. It was observed that a significantly higher proportion of mice developed detectable levels of Immunoglobulin G when vaccinated with liposome-encapsulated DNA as compared to the mice vaccinated with naked DNA [132]. Plasmid-DNA encapsulated liposomes encoding the heat shock protein 65 (HSP65) were found to elicit protective immune responses thereby reducing the fungal burden in pulmonary paracoccidiomycosis [133]. Liu and his co-workers developed a liposome-encapsulated influenza DNA vaccine using the plasmid pcDNA 3.1(+), a eukaryotic expression vector. In vitro cell line studies and in vivo studies on C57BL/6 mice demonstrated the expression of the M1 gene of the influenza A virus. Further, induction of both humoral and cellular immune responses and enhanced IFN-γ production were observed. Thus, the study concluded that oral vaccination with liposome-encapsulated DNA vaccine was capable of protecting the mice against respiratory challenge infection [134]. DNA vaccines against human immunodeficiency virus type 1 were tested in humans for the first time in 1998 [135]. Multiple human clinical trials are currently underway to investigate the potential immunization options for various infectious diseases including those caused by the human papillomavirus (HPV), human immunodeficiency virus (HIV), hepatitis B virus (HBV), and SARS CoV-2. (Table 2).

## 9. Conclusions and Future Perspective

The development of efficacious vaccines with a good safety and tolerability profile in the past has triggered extensive research on novel strategies to improve the shortcomings of current immunization approaches, which include poor immunogenicity and enhanced exposure of susceptible antigens to biochemical degradation. Lipid-based nanocarriers act as an ideal vaccine delivery carrier system. However, as the product progresses from the bench to the bedside, it is a dire need that the method which has been developed based on small-laboratory scale research should meet the production requirements on a larger scale without changing any characteristics of lipid-based vaccines. The minimization of batch-to-batch variability in terms of its physicochemical characteristics such as size, morphology, charge, lamellarity, and antigen release profile are crucial requirements. Higher demand for vaccines in response to a sudden surge of infectious diseases requires a higher throughput with respect to continuous manufacturing and storage requirements. The recently approved liposomal-based vaccines have overcome the above challenges making it a promising platform technology from the pharmaceutical perspective. It is therefore predictable that lipid-based vaccine nanocarriers will be extensively applied in the near future with major success paving way for significant improvements in modern vaccine development.

## Figures and Tables

**Figure 1 vaccines-11-00661-f001:**
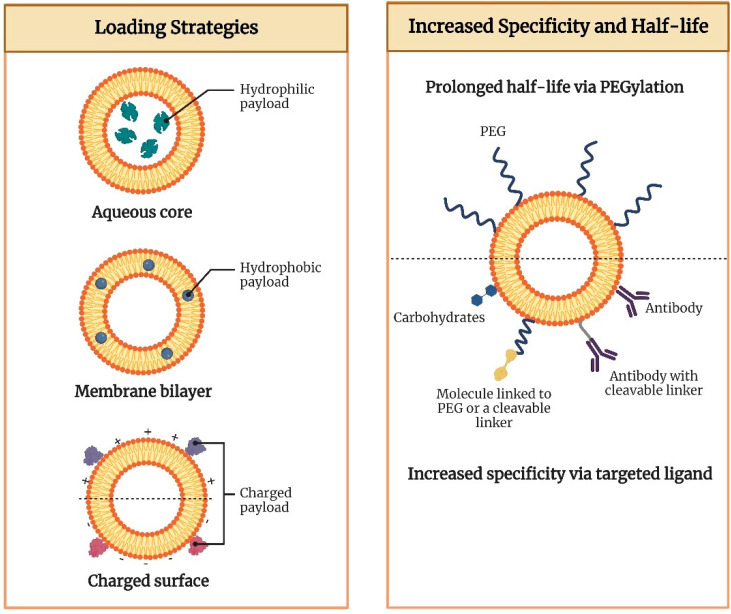
Antigen loading strategies and the potential ways to increase specificity and half-life of lipid nanocarriers.

**Figure 2 vaccines-11-00661-f002:**
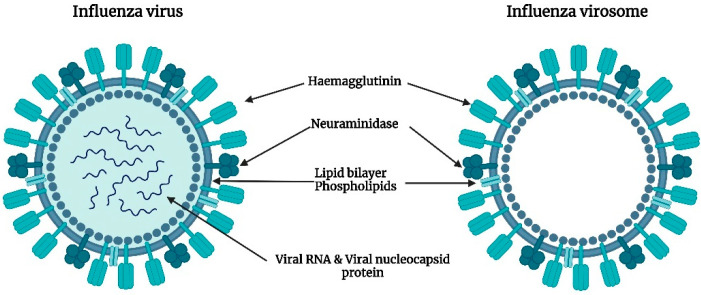
Representative image of an influenza virosome devoid of genetic information that mimics native influenza viruses in terms of cellular uptake and membrane fusion.

**Figure 3 vaccines-11-00661-f003:**
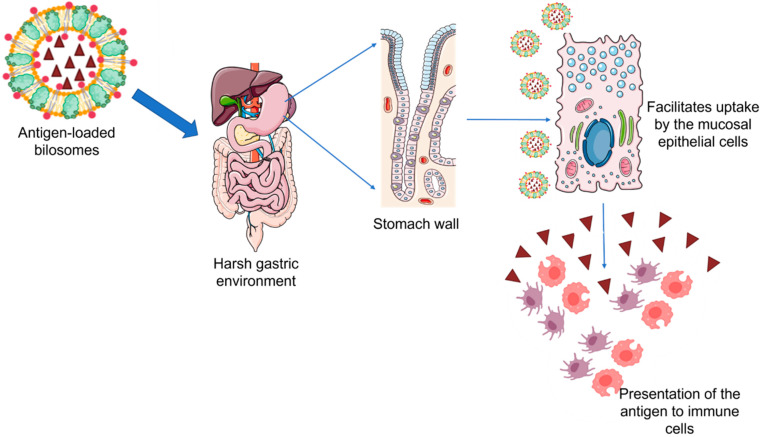
Antigen-loaded bilosomes escape degradation when exposed to the harsh gastric environment facilitating the uptake of antigen through the mucosal cells and its subsequent presentation to the macrophages and dendritic cells.

**Figure 4 vaccines-11-00661-f004:**
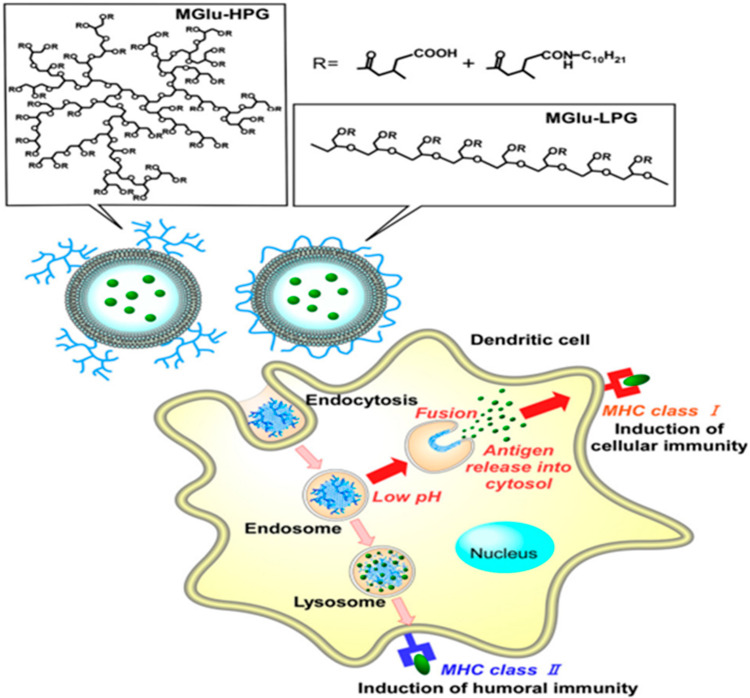
Mechanism of induction of cellular immunity by liposomes modified with poly(glycidol) derivatives. Reprinted with permission from Ref. [45]. 2012, Elsevier.

**Figure 5 vaccines-11-00661-f005:**
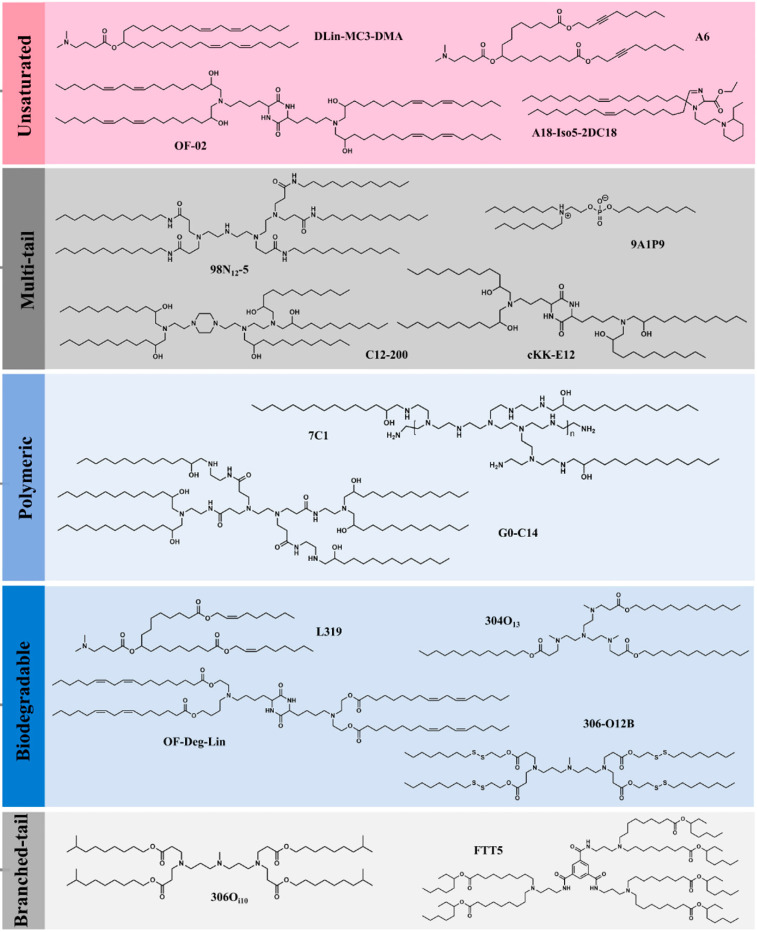
The various structural classes of ionizable lipids utilized in the preparation of LNPs. Reprinted with permission from Ref. [62]. 2021, Springer Nature.

**Figure 6 vaccines-11-00661-f006:**
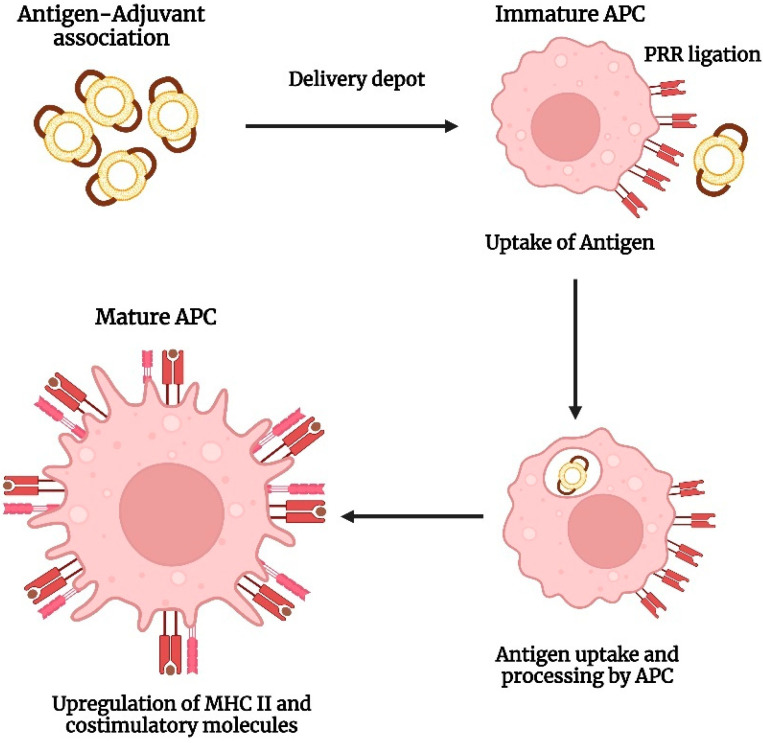
Liposomal adjuvant mediated formation of an antigen-adjuvant depot that promotes the delivery of antigen and subsequent uptake by the APCs.

**Figure 7 vaccines-11-00661-f007:**
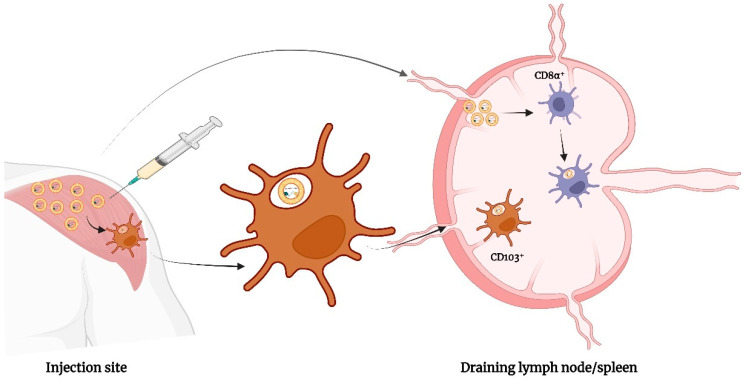
Self-drainage of CAF09 to the dynamic lymph nodes upon intraperitoneal immunization enables direct interaction with CD8α^+^ dendritic cells that possess the ability to cross-present the antigen. Simultaneously, CD103^+^ dendritic cells at the site of injection take up the vaccine particles by phagocytosis that subsequently migrate to the draining lymph nodes.

**Figure 8 vaccines-11-00661-f008:**
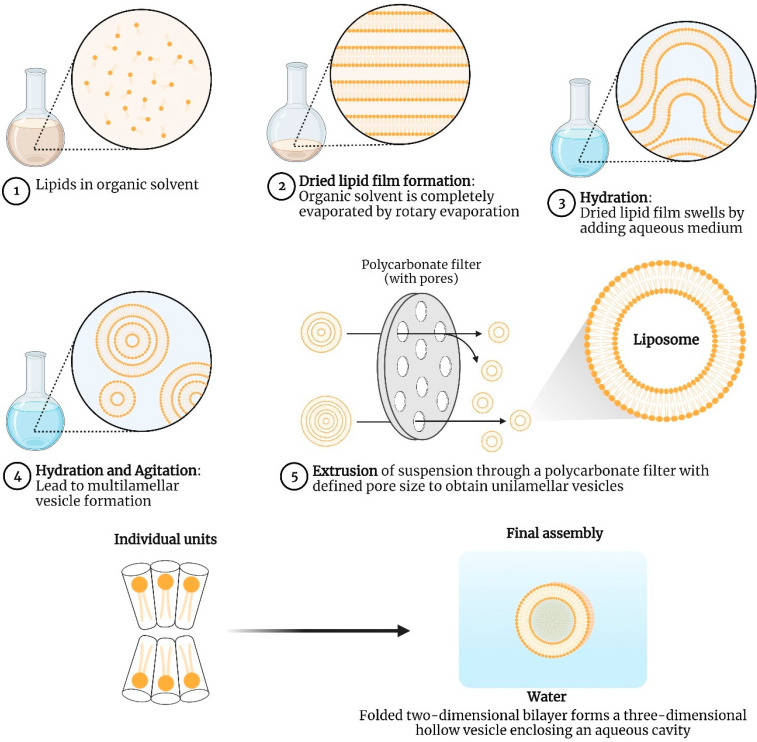
Thin film hydration method for liposome preparation.

**Figure 9 vaccines-11-00661-f009:**
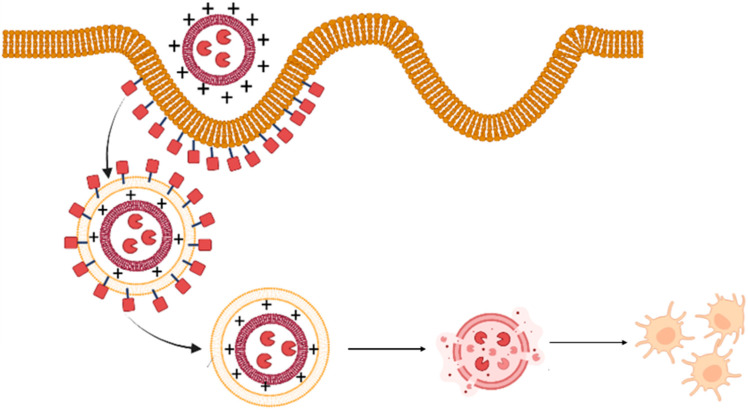
Internalization of cationic liposomes by clathrin-mediated endocytosis causes intracellular delivery of the antigen and subsequent elicitation of the immune response.

**Figure 10 vaccines-11-00661-f010:**
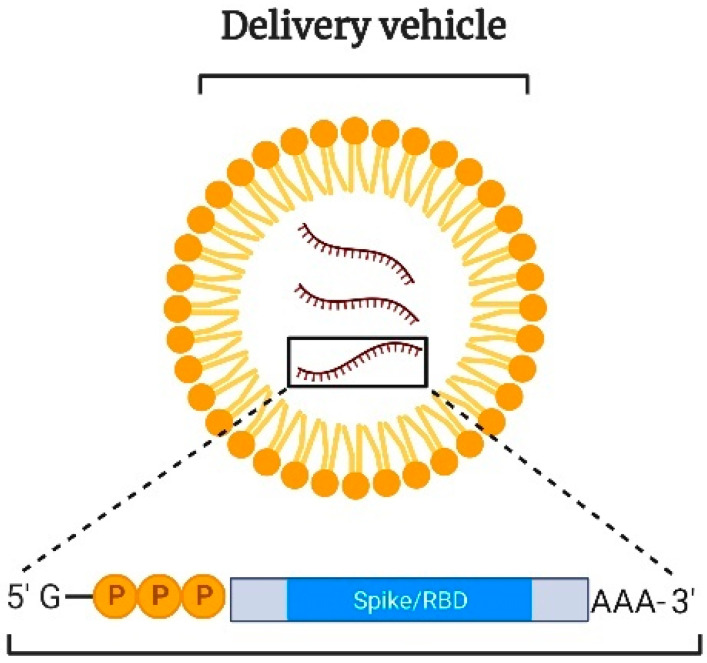
Representative lipid nanoparticle encapsulated mRNA vaccines encoding Spike protein or Receptor Binding Domain (RBD).

**Table 1 vaccines-11-00661-t001:** Clinical trials of lipid-based–mRNA vaccines (https://clinicaltrials.gov/, accessed on 13 February 2023) [125].

Clinical Trial No.	Phase	Name	Route of Administration	Diseases	Antigens
NCT03076385	I	mRNA-1440	IM	Influenza H10N8	Hemagglutinin
NCT03345043	I	mRNA-1851	IM	Influenza H7N9	Hemagglutinin
NCT04064905	I	mRNA-1893	IM	Zika virus	Pre-membrane and envelope glycoproteins
NCT04528719	I	mRNA-1345	IM	Respiratory syncytial virus	F glycoprotein
NCT03392389	I	mRNA-1653	IM	Metapneumovirus and parainfluenza virus type 3 (MPV/PIV3)	MPV and PIV3 F glycoproteins
NCT04232280	I	mRNA-1647	IM	Cytomegalovirus	Pentameric complex and B glycoprotein
NCT03325075	II	mRNA-1388	IM	Chikungunya virus	Chikungunya virus antigens
NCT03713086	I	CV7202	IM	Rabies virus	G glycoprotein
NCT03948763	I	mRNA-5671/V941	IM	Non-small-cell lung cancer/colorectal cancer/pancreatic adenocarcinoma	KRAS antigens
NCT03897881	I	mRNA-4157	IM	Melanoma	Personalized neoantigens
NCT03480152	I/II	mRNA-4650	IM	Gastrointestinal cancer	Personalized neoantigens
NCT02410733	I	FixVac	IV	Melanoma	NY-ESO-1/tyrosinase/MAGE-A3/TPTE antigen
NCT02316457	I	TNBC- MERIT	IV	Triple- negative breast cancer	Personalized neoantigens
NCT03418480	I	HARE-40	ID	HPV-positive cancers	HPV oncoproteins E6 and E7
NCT03815058	II	RO7198457	IV	Melanoma	Personalized neoantigens
NCT04163094	I	W_ova1	IV	Ovarian cancer	Ovarian cancer antigens
NCT03739931	I	mRNA Vaccine	Intra-tumoral (IT)	Solid tumors and Lymphomas	Ligand for OX40 receptor associated with tumor necrosis factor receptor superfamily (OX40L)
NCT03323398	I/II	mRNA Vaccine	IT	Solid tumors, lymphomas and ovarian cancer	OX40L
NCT04283461	I	mRNA Vaccine	IM	COVID-19	S protein
NCT03767270	I	mRNA Vaccine	IV	Ornithine Transcarbamylase (OTC) deficiency	Ornithine transcarbamylase (OTC)
NCT04064905	I	mRNA Vaccine	IM	Zika	pre-membrane protein (prM) and envelope protein (E)
NCT03382405	I	mRNA-1647l, mRNA-1443	IM	CMV infection	Pentamer and T-cell antigen
NCT03713086	I	unmodified mRNA vaccine: CV7202	IM	Rabies	Rabies virus glycoprotein (RABV-G)-mRNA vaccine
NCT03014089	I	mRNA-1325	IM	Zika	Viral antigens

**Table 2 vaccines-11-00661-t002:** Lipid-based DNA vaccines are under clinical trials (https://clinicaltrials.gov/, accessed on 13 February 2023) [125].

Disease	Route of Administration	Name	Clinical Trial No.
HIV Infection	IM	Env-C Plasmid DNA	NCT04826094
HSV-2	Particle Mediated Epidermal Delivery/PowderMed ND10 delivery system	pPJV7630 with pPJV2012	NCT00310271
HIV Infections	injection/electroporation (EP)	HIV-1-Gag	NCT03560258
SARS-CoV-2	EP	GX-19N	NCT04715997
SARS-CoV-2	IM	Covigenix VAX-001	NCT04591184
HSV-2	IM	VCL-HB01	NCT02030301
Hantaan virus (HTNV), Puumala virus (PUUV)	Intradermal Delivery (ID) and IM delivery using TriGrid Delivery System	HTNV/PUUV DNA vaccine	NCT03718130
Chronic Hepatitis C Virus	Electroporation-Mediated Plasmid DNA Vaccine Therapy	HCV DNA Vaccine INO-8000	NCT02772003
HPV16 Positive Cervical Neoplasia	IM using the Trigrid Delivery system	pNGVL4aCRTE6E7L2	NCT04131413

## Data Availability

Not applicable.

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
