# Peer review of "Emerging Trends in Lipid-Based Vaccine Delivery: A Special Focus on Developmental Strategies, Fabrication Methods, and Applications"

_vaccines, 2023, doi:10.3390/vaccines11030661_

Round 1
Reviewer 1 Report
Reviewer comments to the authors:
The current manuscript attempts to review emerging trends in liposomal vaccine delivery with a particular focus on their development, fabrication, and applications. The authors briefly discussed various liposomal/vesicular systems for the delivery of vaccines. After carefully reading through, I am of the impression that the manuscript is in bits and pieces and misses critical insights and is repeatedly misleading. However, the topic is currently relevant and could interest readers. Therefore, I recommend a major revision to the manuscript before further consideration for publication. Below are my comments:
1. ‘Liposomal vaccine delivery’ in the title is not justified with the content of the manuscript and is misleading. I would suggest using ‘lipid-based vaccine delivery systems’ instead wherever relevant in the manuscript. There is very little review of the actual liposome vaccine literature.
2. Section 3: It makes sense to include other lipidic vesicular systems such as transferosomes, and immuno-liposomes. If they are suitable for vaccine delivery/ as adjuvants?
It is also essential to discuss the drawbacks of the mentioned liposome-based systems, and what parameters related to drug/antigen are to be considered while selecting a specific liposome-based system.
3. I do not see any mention of ionizable lipids. Why?
4. Section 5: The subsections are vague and do not contain information on the drug/antigen loading techniques with relevant literature examples. What is the industrial scalability of each of these methods?
5. Section 7.2: Are there ‘liposomal’ DNA vaccines? Information is inadequate.
6. Table 1: these are LNP-based trials. Aren’t there any liposome-based clinical trials? The table isn’t consistent with the manuscript title.
7. Section 7.1: The examples/trials cited were LNP-based trials but not liposomes.
Author Response
Reviewer 1.
The current manuscript attempts to review emerging trends in liposomal vaccine delivery with a particular focus on their development, fabrication, and applications. The authors briefly discussed various liposomal/vesicular systems for the delivery of vaccines. After carefully reading through, I am of the impression that the manuscript is in bits and pieces and misses critical insights and is repeatedly misleading. However, the topic is currently relevant and could interest readers. Therefore, I recommend a major revision to the manuscript before further consideration for publication. Below are my comments:
- ‘Liposomal vaccine delivery’ in the title is not justified with the content of the manuscript and is misleading. I would suggest using ‘lipid-based vaccine delivery systems’ instead wherever relevant in the manuscript. There is very little review of the actual liposome vaccine literature.
Ans: As per the reviewer’s suggestion, the title of the manuscript has been modified. Further, various lipid-based nanocarriers have been included in the revised manuscript and highlighted in green color.
- Section 3: It makes sense to include other lipidic vesicular systems such as transferosomes, and immuno-liposomes. If they are suitable for vaccine delivery/ as adjuvants?
It is also essential to discuss the drawbacks of the mentioned liposome-based systems, and what parameters related to drug/antigen are to be considered while selecting a specific liposome-based system.
Ans: As per the reviewer’s suggestion, suggested lipidic vesicular systems have been included in the revised manuscript and highlighted in green color. The drawbacks/ limitations of the liposome-based system have also been in the revised manuscript.
- I do not see any mention of ionizable lipids. Why?
Ans: As per the reviewer’s suggestion, ionizable lipids have been included in the revised manuscript with suitable figures and highlighted in green color.
- Section 5: The subsections are vague and do not contain information on the drug/antigen loading techniques with relevant literature examples. What is the industrial scalability of each of these methods?
Ans: As per the reviewer’s suggestion, relevant examples have been included in mentioned subsection and highlighted in green color. Further, the industrial scalability of the method has also been highlighted in the revised manuscript.
- Section 7.2: Are there ‘liposomal’ DNA vaccines? Information is inadequate.
Ans: As per the reviewer’s suggestion, adequate information has been included in the revised manuscript.
- Table 1: these are LNP-based trials. Aren’t there any liposome-based clinical trials? The table isn’t consistent with the manuscript title.
Ans: These are lipid-based clinical trials. In some clinical trials, investigators have mentioned liposomes-based vaccines and in some clinical trial, they have mentioned lipid-based systems. However, the composition of lipid-based trials in reported literature is the same as the liposomes.
Furthermore, to keep the consistency with the revised title and content of the manuscript. We kept the title of the table as it is.
- Section 7.1: The examples/trials cited were LNP-based trials but not liposomes.
Ans: Required changes have been made in the revised manuscript.
Reviewer 2 Report
The manuscript “Emerging Trends in Liposomal Vaccine Delivery: A Special Focus on Developmental Strategies, Fabrication Methods and Applications” by Bharathi K. et al. is devoted to various types of liposomes used in a vaccine delivery and the mechanisms by which liposomes activate immune response. Further, the authors try to give a comprehensive analysis of emerging trends and recent developmental strategies on liposome-based vaccine carriers and their role as vaccine adjuvants. This topic is relevant in today's world. However, the authors failed to realize the idea of the review. They do not provide sufficient references to the primary sources of the material they describe. The text is poorly written and does not reflect the essence of the stated topic. The text omits a lot of information known on the subject to date. The provided manuscript does not correspond to the concept of a review article. The authors should refine the material of the article a lot so that it can be published. In the form provided, the review of the Bharathi K. et al. is not suitable for publication.
Author Response
Reviewer 2.
The manuscript “Emerging Trends in Liposomal Vaccine Delivery: A Special Focus on Developmental Strategies, Fabrication Methods and Applications” by Bharathi K. et al. is devoted to various types of liposomes used in a vaccine delivery and the mechanisms by which liposomes activate immune response. Further, the authors try to give a comprehensive analysis of emerging trends and recent developmental strategies on liposome-based vaccine carriers and their role as vaccine adjuvants. This topic is relevant in today's world. However, the authors failed to realize the idea of the review. They do not provide sufficient references to the primary sources of the material they describe. The text is poorly written and does not reflect the essence of the stated topic. The text omits a lot of information known on the subject to date. The provided manuscript does not correspond to the concept of a review article. The authors should refine the material of the article a lot so that it can be published. In the form provided, the review of the Bharathi K. et al. is not suitable for publication.
Ans: As per the reviewer’s suggestion, more references have been included in the revised manuscript. The text of the whole manuscript has also been verified for content improvement. Further, additional information related to the subject of review has been included in the revised manuscript and highlighted in green colour. Now, we believe that the revised manuscript would full fill the reviewer’s requirement.
Reviewer 3 Report
In this manuscript, Kommineni and co-workers did a great job reviewing liposomes and liposome-based nanocarriers used for vaccine delivery purposes. Detailed background covering the basics of liposome carriers has been reviewed well, including the vesicle types, preparation methods, and their immunostimulatory activation mechanism, etc. Finally, examples utilizing liposomes as adjuvants have been discussed, along with recent mRNA vaccines. Liposomes have been proven as effective drug or vaccine delivery vehicles due to their excellent properties. This manuscript is well-written and easy to follow. The reviewer finds it can be a nice addition to the field and supports its publication in Vaccines in its current form.
Author Response
Reviewer 3.
In this manuscript, Kommineni and co-workers did a great job reviewing liposomes and liposome-based nanocarriers used for vaccine delivery purposes. Detailed background covering the basics of liposome carriers has been reviewed well, including the vesicle types, preparation methods, and their immunostimulatory activation mechanism, etc. Finally, examples utilizing liposomes as adjuvants have been discussed, along with recent mRNA vaccines. Liposomes have been proven as effective drug or vaccine delivery vehicles due to their excellent properties. This manuscript is well-written and easy to follow. The reviewer finds it can be a nice addition to the field and supports its publication in Vaccines in its current form.
Ans: We are thankful to the reviewer for the motivational statements.
We appreciate the suggestions of the reviewers to our original submission and thank you for allowing us to revise the same.
Reviewer 4 Report
1. Drawbacks of liposome-based systems are very briefly introduced. For a comprehensive review such as this, section 3.1 does not contain enough information. Details on the current landscape of liposomes (in the first paragraph of introduction) are more fitting to this section. Overall, section 3.1 is not structured or reads the way subsequent sections do. This should be rectified.
2. Parameters related to drug/antigen are to be considered while selecting a specific liposome-based system are not addressed by the authors.
3. Table 1 is not cited in the text.
4. â—¦C should be written as °C.
5. Given the large number of abbreviations in the manuscript, it may be helpful to the readers to refer to a list of important abbreviations.
6. Please ensure correct use of hyphens, dashes for negative signs and en dashes wherever necessary in the manuscript.
7. Please ensure consistent use of sentence case for all headings.
Author Response
- Drawbacks of liposome-based systems are very briefly introduced. For a comprehensive review such as this, section 3.1 does not contain enough information. Details on the current landscape of liposomes (in the first paragraph of introduction) are more fitting to this section. Overall, section 3.1 is not structured or reads the way subsequent sections do. This should be rectified.
Ans: As per the reviewer’s suggestion, section 3.1 has been improved in the revised manuscript and highlighted in yellow color.
- Parameters related to drug/antigen are to be considered while selecting a specific liposome-based system are not addressed by the authors.
Ans: As per the reviewer’s suggestion, the parameters such as log P, polarity, and surface adsorption of antigens have been included in the revised manuscript and highlighted in yellow color (lines 148-163). Further, we would like to bring to the reviewer’s notice that the efficiency of loading mostly depends on the parameters that are related to the carriers. This aspect has been discussed in detail in section 4.
- Table 1 is not cited in the text.
Ans: As per the reviewer’s suggestion, table 1 has been cited in the text of the revised manuscript.
- â—¦C should be written as °C.
Ans: Required changes have been made in the revised manuscript.
- Given the large number of abbreviations in the manuscript, it may be helpful to the readers to refer to a list of important abbreviations.
Ans: As suggested by the reviewer, a list of important abbreviations has been included in the revised manuscript.
- Please ensure correct use of hyphens, dashes for negative signs and en dashes wherever necessary in the manuscript.
Ans: Required changes have been made in the revised manuscript.
- Please ensure consistent use of sentence case for all headings.
Ans: As per the reviewer’s suggestion, all the headings have been verified for consistency.